# The Role of Alarmins in Osteoarthritis Pathogenesis: HMGB1, S100B and IL-33

**DOI:** 10.3390/ijms241512143

**Published:** 2023-07-29

**Authors:** Antonino Palumbo, Fabiola Atzeni, Giuseppe Murdaca, Sebastiano Gangemi

**Affiliations:** 1Rheumatology Unit, Department of Experimental and Internal Medicine, University of Messina, 98124 Messina, Italy; antonino.palumbo.95@gmail.com (A.P.); fabiola.atzeni@unime.it (F.A.); 2Department of Internal Medicine, University of Genova, 16132 Genova, Italy; 3IRCCS Ospedale Policlinico San Martino, 16132 Genova, Italy; 4School and Operative Unit of Allergy and Clinical Immunology, Department of Clinical and Experimental Medicine, University of Messina, 98125 Messina, Italy; gangemis@unime.it

**Keywords:** osteoarthritis, alarmins, DAMPs, cytokines, HMGB1, S100B, IL-33

## Abstract

Osteoarthritis (OA) is a multifactorial disease in which genetics, aging, obesity, and trauma are well-known risk factors. It is the most prevalent joint disease and the largest disability problem worldwide. Recent findings have described the role of damage-associated molecular patterns (DAMPs) in the course of the disease. In particular, alarmins such as HMGB1, IL-33, and S100B, appear implicated in enhancing articular inflammation and favouring a catabolic switch in OA chondrocytes. The aims of this review are to clarify the molecular signalling of these three molecules in OA pathogenesis, to identify their possible use as staging biomarkers, and, most importantly, to find out whether they could be possible therapeutic targets. Osteoarthritic cartilage expresses increased levels of all three alarmins. HMGB1, in particular, is the most studied alarmin with increased levels in cartilage, synovium, and synovial fluid of OA patients. High levels of HMGB1 in synovial fluid of OA joints are positively correlated with radiological and clinical severity. Counteracting HMGB1 strategies have revealed improving results in articular cells from OA patients and in OA animal models. Therefore, drugs against this alarmin, such as anti-HMGB1 antibodies, could be new treatment possibilities that can modify the disease course since available medications only alleviate symptoms.

## 1. Introduction

Osteoarthritis (OA) is the most common subtype of arthritis affecting older people [1,2,3]; it is a chronic and degenerative joint disorder characterised by hyaline cartilage damage, subchondral bone remodelling, chondrophyte and osteophyte formation, and synovitis [4,5,6,7]. Different risk factors concur in the pathogenesis and the most common are aging, obesity, trauma, female sex, and genetics [8,9,10,11]. The first event in the course of the disease is hyaline cartilage degeneration. Cartilage is an avascular and nerveless tissue formed by chondrocytes, the only cell type, and extracellular matrix, secreted by chondrocytes themselves and high in proteoglycans and collagen. Chondrocytes are implicated in cartilage homeostasis; indeed, any insult affecting them, such as mechanical trauma, abuse, and/or repetitive loads, or hypoxia, results in alterations of the extracellular matrix [12]. After a joint injury occurs, a significant quantity of chondrocytes die along the surface of cartilage, and adjacent cells are driven by different mediators to switch into catabolic molecular activities, so the excess of degradative over reparative processes results in gradual loss of articular cartilage and joint space narrowing [13,14,15].

Despite the chronic and degenerative definition of OA, the pathogenesis is driven by different inflammatory mediators, such as cytokines, chemokines, damage-associated molecular patterns (DAMPs), matrix proteinases (MMPs and ADAMTS), and eicosanoids [15,16,17]. Tumor necrosis factor α (TNF-α) and mostly interleukin 1β (IL-1β) are involved in catabolic enzymes, nitric oxide (NO), and prostaglandins synthesis and in suppressing extracellular matrix components production by chondrocytes [14,16,18].

OA most common symptoms are joint pain, effusion, crepitus, reduced range of motion, and stiffness [8]. It carries an important economic burden worldwide because of its large prevalence, chronicity, and progression, which make it a disabling disease [1,19]. For all these reasons, it is relevant to increase knowledge about OA pathogenesis in order to undertake new therapeutic perspectives to control the disease.

## 2. Alarmins: HMGB1, IL-33, S100B

Alarmins are a group of mediators that play a role in the innate inflammatory response. They are so named because they are usually released by cells from injured tissues, and thus are also known as damage-associated molecular patterns (DAMPs). In particular, alarmins are endogenous DAMPs, so it is important to distinguish them from exogenous DAMPs, derived from pathogens, the so-called pathogen-associated molecular patterns (PAMPs) [20,21,22].

### 2.1. HMGB1

High mobility group box 1 (HMGB1) is a nuclear nonhistone DNA binding protein, so named for its rapid mobility on gel electrophoresis [23]. HMGB1 belongs to the high mobility group proteins (HMGs), which are divided into three families: HMGA, HMGB, and HMGN, respectively, distinguished by the AT-hook domain, two boxes of DNA-binding domains, and a nucleosomal binding domain [24,25,26]. The HMGB family includes four proteins: HMGB1, HMGB2, HMGB3, and HMBG4. HMGB1 has three different domains: box A and box B, two DNA-binding motifs, and acid C-terminal tails [23,27,28]. Box B is the active cytokine domain while box A limits the function of the alarmin [27]. HMGB1 can be actively or passively released outside the cell; inflammatory stimulations mediate the translocation from the nucleus to the cytoplasm and then to the extracellular medium, while necrosis and apoptosis are passive modalities of HMGB1 cell release [29,30,31,32,33]. It has different functions depending on its localisation: in the nucleus, HMGB1 promotes nucleosomal stability and supports the bond between transcriptional factors and their DNA target; when released in the extracellular milieu, it chemo-attracts osteoblasts, osteoclasts, and endothelial cells during endochondral ossifications and promotes transendothelial migration of monocytes and amplification of inflammatory response [23,30,34,35]. HMGB1 interacts with multiple receptors: Toll-like receptors (TLR2 and TLR4), C-X-C motif receptor 4 (CXCR4), and receptors for advanced glycation end products (RAGEs), leading to innate immune activation and inflammation amplification. In particular, HMGB1 interactions depend on the status of the three conserved cysteine domains: disulphide HMGB1 (dsHMGB1) interacts with TLR4, inducing cytokines production and inflammation in vitro; all-thiol HMGB1 (atHMGB1) binds to CXCR4 and RAGE, which are involved in chemotaxis and cell recruitment [30,36,37,38,39]. Another form is oxidised HMGB1, which is produced by the oxidation of all three cysteine residues through the continuous production of reactive oxygen species (ROS), and it appears not to generate any inflammatory response or chemo-attractive function but is involved in inflammation resolution [40]. Finally, hyperacetylation of lysine residues in the nuclear localisation sequences (NLSs) is necessary to determine the cytoplasmatic accumulation of HMGB1, as well as HMGB1 pro-inflammatory power acquisition [30,41]. The pro-inflammatory role of HMGB1 is also mediated by its capacity to form complexes with other agents, such as lipopolysaccharide (LPS), IL-1β, C-X-C motif ligand 12 (CXCL12), interferon γ (INF-γ), and CpG-DNA (synthetic DNA molecules formed by a single strand containing a cytosine followed by a guanine) [42,43,44,45]. HMGB1 is able to stimulate different types of cells, such as chondrocytes, monocytes, and lymphocytes. In particular, its pro-inflammatory power depends on cell type and doses. For example, HMGB1-stimulated peripheral blood monocytes release TNF-α, IL-1α, IL-1β, IL-6, IL-8, macrophage inflammatory protein 1 α and β (MIP-1α and MIP-1β) [46].

### 2.2. IL-33

IL-33 belongs to the IL-1 cytokines family, which includes IL-1α, IL-1β, IL-1 receptor antagonist, and IL-18 [47,48]. It is an alarmin released by apoptotic and necrotic cells, epithelial and endothelial stressed cells [49,50]. It is synthesised as pro-IL-33 and then released in the extracellular medium after cleavage. This interleukin binds various receptors and activates different intracellular signalling. In particular, IL-33 induces NF-κB activation by binding the T1/ST2 (suppressor of tumorigenity) receptor (ST2L), which belongs to the Toll-like/IL-1 receptor family; furthermore, IL-33 enhances other TLRs pathways, especially TLR2 and TLR4. Additionally, the ST2 receptor has three isoforms produced by the same transcript through differential splicing. ST2L is the transmembrane long form expressed by T helper 2 cells (TH2) and mast cells, which mediates the effect of IL-33 in these cell subtypes. Soluble ST2 (sST2) acts as a decoy receptor by reducing IL-33 interactions with other forms of ST2 receptors, attenuating inflammatory response [51]. ST2V is expressed in the stomach, small and large intestine, and spleen [52]. As seen in other alarmins, IL-33 works as a booster of the inflammatory response, such as LPS in its TLR4 binding—IL-33 increases the synthesis of molecules involved in downstream signalling, favouring activation and pro-inflammatory cytokines and chemokines secretion [53]. This interleukin mediates several functions: it facilitates the polarisation of macrophages towards the M1 pro-inflammatory phenotype, enhances adhesion in human basophils and eosinophils, stimulates inflammatory cytokines production by mast cells, and augments IL-5 and IL-13 secretion by TH2 cells and IFN-γ secretion by natural killer (NK) cells [47,54,55,56,57].

### 2.3. S100B

S100B is a calcium-binding protein belonging to the S100 family formed by 24 different members participating as intracellular or extracellular regulators. This alarmin mediates several processes inside cells, such as apoptosis, cell proliferation, differentiation, metabolism, and Ca^2+^ homeostasis. It is released in the extracellular medium and acts as a DAMP, activating different receptors, such as RAGE and TLR4. RAGE and TLR4, when activated by S100B, promote their intracellular signalling of extracellular signal-regulated kinase-mitogen-activated protein kinase (ERK-MAPK) and nuclear factor kappa-light-chain-enhancer of activated B cells (NF-κB) [24]. S100B enhances inflammatory responses in lymphocytes, macrophages, cardiac cells, and endothelial and vascular smooth muscle cells [58,59].

## 3. Aim of the Study

This review offers a description of the association of HMGB1, IL-33, and S100B with osteoarthritis, clarifying their signalling (a), their role in the pathogenesis of OA (b), identifying possibilities of using them as biomarkers of the disease (c) and its severity (d), and offering future therapeutic perspectives (e). A considerable number of papers published on the relationships between these three alarmins and osteoarthritis, and the general coherence in their results were a source of inspiration for this paper.

## 4. Search Strategy

The search was performed in Pubmed and included articles published between 2005 and 2022. Only original articles written in English were considered. Review papers and articles outside the scope of this review were not taken into consideration. Each article was screened for relevance and coherence with the topic of the study. The articles were searched as follows: “HMGB1” AND “osteoarthritis”; “S100 b protein human” AND “osteoarthritis”; “Interleukin-33” AND “osteoarthritis”.

## 5. Results and Discussion

### 5.1. Role of HMGB1 in Osteoarthritis

In the last years, increasing attention has been directed to the role of HMGB1 in OA initiation and progression. Different papers have confirmed that OA chondrocytes show an augmented expression of HMGB1 compared with normal cartilage [60,61,62,63]. Amin and colleagues have reported an increased expression of HMGB1 in the deep layers of OA articular cartilage compared with normal cartilage [63]. Additionally, it has been found that HMGB1 levels in the tidemark with subchondral bone are positively correlated with the histopathological grade of OA [60,61]. HMGB1 expression in OA cartilage is prevalent in the cytoplasm and extracellular medium rather than in normal cells, whose HMGB1 positivity is predominantly nucleosomal [60,61,63,64].

HMGB1 is accumulated from necrotic or apoptotic cells and is actively released by OA chondrocytes [60,63,65]. DAMPs, derived from injured cartilage, can induce translocation of HMGB1 from the nucleus to the cytoplasm and then in the extracellular space. Hwang and colleagues have demonstrated that 29-kDa amino-terminal fibronectin fragment (29-kDa FN-f), a DAMP derived from the extracellular matrix and found in synovial fluid of OA patients, increases the extracellular release of HMGB1 by OA chondrocytes. In addition, 29-kDaFN-f interferes with the autophagy mechanism, necessary for the homeostatic status of articular cartilage. Specifically, autophagy is inhibited by elevated phosphorylation of mTOR and by complex formation of Beclin-1/Bcl-2, favoured over Beclin-1/HMBG1, leading, in turn, to autophagy [65].

Different papers have also described the influence of some cytokines on HMGB1 expression in OA chondrocytes. In vitro, tests have confirmed that OA chondrocytes, incubated with IL-1β or TNF-α, increase HMGB1 expression and nucleus to cytoplasm translocation (Figure 1) [60,61,66,67]. Additionally, IL-1β appears to reduce chondrocyte viability in a dose-dependent manner [67]. For all these reasons, many scholars incubate normal or OA chondrocytes with IL-1β or LPS to reproduce in vitro models of OA and perform tests.

HMGB1 has demonstrated a pro-inflammatory role in OA—in particular, it is able to induce augmented expression of different mediators. Several studies have reported that chondrocytes from OA-affected patients, incubated with recombinant dsHMGB1, were induced to increase their chemokines, interleukins, and inducible nitric oxide synthase (iNOs) productions, while anabolic synthesis of collagen type II alpha 1 chain (COL2A1) was decreased [63,68]. Terada and colleagues have also shown that HMGB1 is able to induce OA chondrocyte augmented release of IL-1β and TNF-α (Figure 2) [60]. Therefore, available evidence underlines the interrelationship between IL-1β and HMGB1 since in vitro studies have demonstrated the ability of one to increase the expression of the other. Additionally, Fu and colleagues have reported that overexpression of HMGB1 A box, the inhibitory domain of HMGB1, in IL-1β-stimulated chondrocytes significantly reduces levels of HMGB1 as well as degradative enzymes and inflammatory mediators [66].

However, it was largely demonstrated that alarmins do not act alone but in a synergic way with other factors: in particular, HMGB1 increases the IL-1β-induced synthesis of matrix metalloproteinases (MMPs), a disintegrin and metalloproteinse (ADAM), and iNOS by normal human chondrocytes; FN-f shows similar results but with a minor synergic effect on IL-1β stimulated chondrocytes [69]. Moreover, various researchers agree that HMGB1 may act on chondrocytes and other cell types in autocrine and paracrine ways to enhance and spread sterile inflammatory response (Figure 1) [60,62,63].

Finally, it is important to underline that OA chondrocytes have shown increased expression of RAGE and TLR4, which are HMGB1 receptors [60,64]. Apoptosis of chondrocytes results in increased secretion of HMGB1 while senescence induces the accumulation of advanced glycation end products (AGEs); therefore, it is possible to assume that the number of AGEs and HMGB1 in cartilage may favour the interaction with RAGE, thus increasing the secretion of pro-inflammatory cytokines and contributing to cartilage degeneration, further confirming its role in OA [62].

Nevertheless, the relationship between HMGB1 and cartilage damage was largely demonstrated, showing that this alarmin is likely to be also involved in tissue repair mechanisms. Wagner and colleagues have reported a chemo-attractive role of HMGB1 on chondrogenic progenitor cells (CPCs) [64]. Heinola and colleagues have described that OA progression is related to an intensification of extracellular levels of HMGB1 in the border between cartilage and bone in animal models [61]. However, further studies are needed to demonstrate the role of recruited CPCs in OA pathogenesis and their possible role in subchondral bone sclerosis and/or osteophyte formation.

The osteoarthritis pathogenesis cascade starts from an injury afflicting chondrocytes, but the activated inflammatory responses then involve all joint tissues: subchondral bone, synovium, synovial fluid, menisci, and ligaments.

It was described that HMGB1 expression is also augmented in OA synovial membrane compared with the normal membrane [67,70,71]. The works of Garcìa-Arnandis et al. and Ke and colleagues have shown that the number of HMGB1-positive OA synoviocytes was higher than that in controls, and HMGB1 positivity was found mostly in the cytoplasm and in the extracellular medium rather than in the nuclei; however, the results of the first paper have not reached statistical significance [71,72]. Sun and colleagues have reported similar evidence: they have found augmented messenger RNA (mRNA) and protein expressions of HMGB1 and RAGE by synovial tissues from knee OA patients, compared with the control group. HMGB1 positivity was predominantly cytoplasmatic; additionally, the authors demonstrated that HMGB1 and RAGE levels were positively correlated with the X-ray grade of the disease [70]. HMGB1 was also found to increase, same as mRNA and protein expressions, in synovial samples from OA temporomandibular joint compared with patients with a condylar fracture [73].

HMGB1 seems to act on synoviocytes in the same way as on chondrocytes. In fact, different works have confirmed the inflammation-enhancing power of HMGB1 on synovial cells. It is widely described that this alarmin could increase interleukins, chemokines, and MMPs productions by OA synoviocytes, acting in complex with other pro-inflammatory mediators, such as LPS, IL-1α, and IL-1β. Specifically, the HMGB1-IL-1β combination induces a higher cytokines production than the HMGB1-LPS complex. These findings were also confirmed by the reversed effects on cytokines productions of OA synoviocytes incubated with detoxified LPS and IL-1 receptor antagonists, indicating the TLR4 and IL-1RI pathways involvement, respectively [71,74,75].

Hence, HMGB1 has been implicated in OA course acting, alone or in couple with other mediators, on chondrocytes and synoviocytes by binding TLR4 and RAGE receptors. During the last few years, different researchers aimed to elucidate the molecular pattern by which HMGB1 leads to OA. NF-κB signalling is considered a key mechanism in the regulation of transcription in OA.

HMGB1 has been shown to significantly increase MMP-13, ADAMTS-5, IL-1β, and IL-6 synthesis in human synovial fibroblasts from OA temporomandibular joint (TMJ) through NF-κB signalling [76]. It was shown that OA human chondrocytes, stimulated by recombinant dsHMGB1, increased synthesis of NF-κB1 (NF-κB p105 subunit) and NF-κB2 (NF-κB p100 subunit) mRNA, two NF-κB light chain enhancers [63]. Qiu and colleagues have reported an increased phosphorylated-NF-κB/NF-κB ratio in IL-1β-induced chondrocytes [77]. Li et al. have described increased expression and nuclear localisation of p-NF-κB p65 in HMGB1-induced synovial fibroblasts from TMJOA [76]. Garcìa-Arnandis and colleagues showed that HMGB1-stimulated OA synoviocytes increased their transcriptional activity through NF-κB; however, the results did not reach statistical significance. On the other hand, HMGB1-IL-1β complex stimulation deeply potentiated NF-κB activation. Last, they described the capacity of HMGB1, in the presence of IL-1β, to enhance the phosphorylation of ERK1/2, p38, and Akt [71].

Jiang and colleagues have demonstrated that HMGB1 augmented expression and translocation from nucleus to cytoplasm in human chondrocytes incubated with IL-1β is due to bromodomain containing 4 (BRD4) increased expression in cartilage from OA-affected patients, compared with non-OA-affected patients. Additionally, BRD4 expression was positively correlated with histological OARSI score, and this finding may partially explain the gradually increased expression of HMGB1 with OA progression. This protein belongs to the bromo and extra-terminal (BET) domain family and plays an epigenetic regulatory role because of its capacity to bind acetylated lysine residues of histone tails. In fact, researchers have postulated and reported that BRD4 acts by binding the HMGB1 promoter and upstream non-promoter region, inducing gene transcription. They have also demonstrated that JQ1, a BET protein inhibitor, has reversed the pro-inflammatory and catabolic effect of IL-1β on human chondrocytes cultures, reducing mRNA and protein levels of HMGB1, IL-6, TNF-α, and MMPs. Finally, BRD4 seems to play a crucial role in the activation of NF-κB signalling since JQ1 reduces the translocation of p65 from the cytoplasm to the nucleus. All these findings have further confirmed the role of BRD4 in OA [78].

The focus has also been directed to the influence of long noncoding RNA (lncRNA) and miRNAs towards HMGB1 expression.

MiRNAs are endogenous noncoding RNAs involved in the regulation of gene expression through the bond with the 3′-untranslated region (3′-UTR) of mRNAs of their target genes.

MCM3AP antisense RNA 1 (MCM3AP-AS1) is a long noncoding RNA primarily described in hepatocellular carcinoma. Gao and colleagues have reported increased expression of this RNA in OA patients compared with controls. They have demonstrated that MCM3AP-AS1 acts by binding miR-142-3p, a miRNA that may improve OA by targeting HMGB1, thus reducing apoptosis and inflammation. In fact, OA chondrocytes incubated with miR-142-3p have shown decreased mRNA and protein levels of HMGB1, while overexpression of MCM3AP-AS1 has led to augmented expression of HMGB1 and has been able to reduce miR-142-3p overexpression effects. Finally, the researchers have reported that LPS-mediated chondrocytes reveal upregulated MCM3AP-AS1 and HMGB1 mRNA and downregulated miR-142-3p in a dose-dependent manner [79].

MiR-140-5p, such as miR-142-3p, confirmed improved effects on OA, reducing apoptosis, pro-inflammatory factors, and matrix metalloproteinases synthesis in IL-1β-induced chondrocytes. It acts by targeting the 3′-UTR of HMGB1, and low levels of miR-140-5p correspond to the high expression of HMGB1 found in OA tissues and IL-1β-induced chondrocytes. Wang et al. have reported that miR-140-5p interferes with PI3K/AKT, a demonstrated downstream pathway of HMGB1; this miRNA has reduced levels of phosphorylated PI3K and phosphorylated AKT (Ser473), as well as HMGB1, in IL-1β-mediated chondrocytes. Finally, miR-140-5p overexpression has revealed reduced levels of MMP-1, MMP-3, TNF-α, IL-6, and apoptosis rate, and increased cell viability [80].

MiR-129-5p is another miRNA with described ability to relieve OA alterations. In vitro tests have shown that it targets 3′-UTR of HMGB1 mRNA, reducing the alarmin release and alleviating inflammatory responses and apoptosis of IL-1β-incubated chondrocytes. In fact, IL-1β-mediated chondrocytes incubated with exosomes poor in miR-129-5p have shown increased HMGB1, TLR4, phosphorylated NF-κB, cyclooxygenase 2 (COX2), iNOS, and MMP13 levels and apoptosis rate, while collagen 2 was downregulated. On the other side, the same cells incubated with exosomes rich in miR-129-5p have revealed inverse results. As in the previous description of miRNAs, miR-129-5p decreased in OA patients’ joint fluid and IL-1β-induced chondrocytes and was negatively correlated with HMGB1 levels [77].

Plasmacytoma variant translocation 1 (PVT1) is another long noncoding RNA involved in OA pathogenesis. Meng and colleagues have reported higher PVT1 levels in the serum of OA patients and LPS-stimulated human normal chondrocytes C28/I2 than in the controls. On the other side, the miR-93-5p expression has shown a negative correlation with PVT1, with lower levels in the serum of OA patients and LPS-stimulated C28/I2. The researchers have demonstrated that PVT1 is implicated in OA pathogenesis since PVT1 downregulation, through small interference targeting PVT1 (si-PVT1), relieves cytokines secretion, collagen degradation, and apoptosis rate in LPS-incubated C28/I2. Furthermore, decreased PVT1 expression increases miR-93-5p levels in vitro. MiR-93-5p has been revealed to be a target for PVT1 and has shown a suppressive target activity towards HMGB1. Finally, they have elucidated that LPS-mediated chondrocytes show increased expression of p-p65, TLR4, and phosphorylated NF-κappa-B inhibitor α (p-IκB-α), and PVT1 inhibition reverses the increase. Therefore, PVT1 acts in the TLR4/NF-κB pathway targeting miR-93-5p/HMGB1 axis [81].

Lin and colleagues have described the role of miR-107 in OA. They have reported that this miRNA targets the 3′-UTR of HMGB1 mRNA, with improving properties in OA conditions. In fact, upregulation of miR-107 decreases HMGB1 levels in OA chondrocytes. Additionally, the researchers have found that under hyperbaric oxygen treatment, OA chondrocytes significantly increase miR-107 expression, reducing the synthesis and extracellular release of HMGB1, inflammatory receptors, and degradative enzymes. HBO therapy has markedly reduced p38 MAPK, ERK, and jun N-terminal kinase (JNK) phosphorylation. Furthermore, the scholars have demonstrated that HBO treatment also counters the NF-κB pathway, increasing IκB-α protein synthesis that prevents nuclear translocation of the p65 subunit of NF-κB [82].

It is plausible that HMGB1 could play a role in the neovascularisation of OA synovia. In fact, Feng and colleagues have reported that HMGB1 increases vascular endothelial growth factor (VEGF) protein expression in synovial fibroblasts from OA temporomandibular joint in a dose-dependent manner; in addition, the conditioned medium from HMGB1-incubated fibroblasts causes human umbilical vein endothelial cells (HUVEC) to migrate and favour new vessel formation. Therefore, the researchers have found that HMGB1 is able to induce VEGF-dependent neo-angiogenesis, by increasing Erk and JNK phosphorylations, which then augment hypoxia-inducible factor 1 α (HIF-1α) cell expression that positively modulates VEGF expression [83].

It was largely demonstrated that HMGB1 extracellular release is increased in OA, determining accumulation and dispersion in the outside-cell medium. For this reason, scholars have investigated whether synovial fluid levels of HMGB1 would be augmented and correspond to disease severity.

Different papers have shown that HMGB1 expression in synovial fluid from OA patients is increased compared with controls [67,72,84]. Additionally, the levels of this alarmin are positively correlated with the radiographic severity of knee OA evaluated with the Kellgren-Lawrence (KL) severity classification system, after adjusting for confounder elements, such as age, body-mass index (BMI), and sex [84]. Otherwise, Ke and colleagues have reported that HMGB1 synovial fluid levels were higher in OA patients radiologically classified in the KL 2/3 group, compared with those who were placed in the KL 4 group. Nevertheless, synovial fluid HMGB1 levels were found to be positively correlated with clinical findings: pain, synovitis, and daily activities [72]. Therefore, HMGB1 seems to be not only a biomarker of OA but also a severity index of its clinical and radiological features; however, further evidence is needed to clarify the possibility of future use in clinical practice.

HMGB1 levels in synovial fluid were compared to time from articular injury, in order to further investigate its involvement in OA timeline. Aulin and colleagues reported that HMGB1 increased in synovial fluid after a knee trauma as a response to the acute injury and its levels were higher compared with the OA and old-injury patient group; however, statistical significance was not reached. These differences were not as marked when compared with S100A8/A9 levels; in fact, S100A8/A9 expression in the acute injury group was deeply increased compared with the old-injury (317.54 vs. 3.07) and OA group (317.54 vs. 7.35). Additionally, HMGB1 is correlated with cartilage turnover makers [85]. Different results were reached by Ding et al.: they reported that HMGB1 levels in synovial fluids from patients with chronic anterior cruciate ligament (ACL) injury were higher than in the acute group, but these results did not reach statistical significance [86]. Therefore, further studies are needed to completely clarify the timeline of HMGB1 in the OA course. Probably, small amounts of this alarmin, realised after a trauma, initiate a cascade of events, which then are gradually amplified and self-maintained.

Bleeding in the knee is considered a risk factor for OA and heme could be a driving factor for disease development. It has been demonstrated that induction of heme oxygenase-1 (HO-1) in vitro interferes with the production and translocation from the nucleus to the cytoplasm of HMGB1 in IL-1β-mediated synoviocytes; additionally, induction of HO-1 in OA-treated synoviocytes has also reduced synthesis of MMP-1 and MMP-3 [87]. It is not clear how the augmented activity of this enzyme could improve OA. It may act by degrading heme and/or the production of antioxidant molecules, such as bilirubin/biliverdin. Based on these results, it is conceivable to assume that repeated microtraumas could produce microbleeding in the joint that may initiate and promote the progression of the disease.

Various researchers have also hypothesised that there may be different pathogenic mechanisms based on the districts affected by osteoarthritis. Rosenberg and colleagues have reported that OA knee cartilage shows higher expression of HMGB1, S100A8, and RAGE than OA hip, both in terms of protein expression and mRNA [10]. The knee is the joint most frequently affected by osteoarthritis. This evidence might suggest a molecular substrate for this higher prevalence. Indeed, the increased expression of alarmins and their receptors would contribute to the progression of the disease due to their pro-inflammatory and catabolic effect on cartilage. However, it is necessary to clarify the cause of this more vivid sterile inflammation. One reason could involve biomechanical factors and stressing forces that may act more on knee chondrocytes than hip cells, leading to a higher rate of HMGB1 release. Otherwise, the knee and hip could be joints with different intrinsic resistance in weight bearing. Further evidence is needed.

In recent years, numerous pieces of evidence have pointed to a positive effect of antioxidant molecules in the mechanisms of OA. Chrysin is a flavonoid of plant origin; flavonoids are known for their antioxidant properties. It has been shown to reduce cell apoptosis and expression of HMGB1, enzymes, and IL-6 in human OA chondrocytes treated with IL-1β; on the other hand, it is able to increase the synthesis of COL2A1 [88]. Another substance that has been revealed to interfere with the pathogenesis of OA is glycyrrhizin. This is a substance contained in liquorice. In vitro studies have reported that glycyrrhizin reduces the expression of HMGB1, prostaglandin E2 (PGE2), NO, MMPs, and interleukins levels in OA chondrocytes incubated with IL-1β [67].

It is plausible that the initial release of HMGB1, probably from necrotic chondrocytes, initiates different molecular events. In effect, HMGB1, acting in an autocrine and/or paracrine way, could increase cytokines and chemokines realising and amplifying their molecular signalling by forming an ethero-complex. All these events could lead cartilage chondrocytes to switch into catabolic activities, secreting enzymes and reducing extracellular matrix component production. The main evidence of the HMGB1 role in OA is summarised in Table 1.

### 5.2. Role of IL-33 in Osteoarthritis

IL-33 is another DAMP that seems to have a role in OA pathogenesis. It has been reported that OA chondrocytes show an augmented quantity of IL-33 mRNA, in association with augmented levels of IL-37 and other interleukins, expression of receptors, and MMPs, compared with normal human chondrocytes [89,90,91]. Additionally, IL-33 protein expression has revealed a greater amount in osteoarthritic cartilage than in normal controls [68,91]; the same results have been found for IL-37, ST2, TLR2, TLR4, NF-κB, IL-6, TNF-α levels, and numerous degradative enzymes [89,90]. It is possible to assume that IL-33 may have a role in OA development. Mechanical stress is unanimously considered the most impacting factor in OA onset, and for this reason, mRNA of IL-33, MMP1, and MMP13 has been measured and found to be higher in weight-bearing cartilage compared with non-weight-bearing areas. In order to deeply understand if different pathogenesis could define OA according to the type of joint, Rai and colleagues have performed quantitative reverse transcription polymerase chain reaction (qRT-PCR) and immunofluorescence in the two groups of tissues: hip and knee cartilages. They have evidenced that mRNA and protein levels of IL-33, TLR2, TLR4, NF-κB, MMP2, and MMP9 were higher in the OA knee compared with the OA hip, while IL-37 protein expression was higher in the hip than in the knee cartilage [89].

IL-33 has also demonstrated a pro-inflammatory power; in fact, this alarmin, such as rHMGB1 and LPS, has increased mRNA levels of IL-33, TNF-α, IL-6, TLR2, TLR4, MMP2, MMP9, NF-κB, HMGB1, and RAGE in incubated normal human articular chondrocytes (NHAC) and has determined M1 phenotype conversion of mediated macrophages. Differently, IL-37 has revealed an anti-inflammatory effect on stimulated NHAC. In fact, this interleukin has reduced mRNA expression of several inflammatory mediators, receptors, and HMGB1 in treated cells compared with non-treated cells. Finally, IL-37 has interfered with the effect of IL-33 since mRNA levels of TLR2, TLR4, IL-6, TNF-α, NF-κB, MMP2, MMP9, RAGE, and HMGB1 decreased in cells incubated with IL-37 followed by IL-33, LPS or rHMGB1 compared with cells incubated only with IL-33, LPS or HMGB1. Additionally, IL-33 has shown an auto-stimulation effect in incubated NHAC that has been further confirmed by decreased synthesis of its mRNA levels in NHAC incubated with ST2 blocking antibody [89].

Great attention has been directed to the molecular pathway of IL-33; in particular, Li and colleagues have demonstrated that IL-33 is released by chondrocytes after double strand RNAs (dsRNAs) bind TLR3. The researchers have realised a three-step experiment: first, they incubated normal human chondrocytes with supernatant from healthy and injured cartilage lysates and then they compared normal human chondrocytes with supernatant from injured cartilage lysates, with and without RNase A. In the last step, they compared results from human chondrocytes stimulated with supernatant from injured cartilage lysates, knocking down TLR3 and TLR7 with siRNA. They have reported that chondrocytes stimulated with damaged cartilage lysate increase mRNA and protein expression of IL-33, MMP1, and MMP13 and reduce collagen II production. The introduction of RNase A in the lysate reduces this effect, and knocking down TLR3 in stimulated chondrocytes is more effective in reducing IL-33, MMP1, and MMP13 levels and increasing collagen type II expression than silencing TLR7. Additionally, they have described that TLR3 induces IL-33 synthesis through p38 MAPK, which acts upstream of p65, which, in turn, forms a complex with peroxisome proliferator-activated receptor gamma (PPARγ). IL-33 then will induce MMP1 and MMP13 release and reduce type II collagen synthesis. In fact, researchers have shown that IL-33-stimulated chondrocytes increase mRNA and protein levels of these two enzymes. Finally, IL-33 seems to act downstream dsRNA, since human chondrocytes incubated with dsRNA analogue increase MMP1 and MMP13 expression, and this phenomenon has been reverted through IL-33 silencing with IL-33 siRNA. Therefore, IL-33 is probably released in OA chondrocytes through TLR3 signalling, in which dsRNAs derived from necrotic and apoptotic cells bind the receptor, and the cartilage tissue homeostasis switches to a catabolic status (Figure 3) [90]. Besides cartilage, synovial fluid from OA patients has also demonstrated a higher presence of IL-33, MMP1, and MMP13, compared with healthy controls [90]. He and colleagues have reported that IL-33 is increased in synovial fluid of OA patients compared with controls but not in sera [91]. Otherwise, Hong and colleagues have reported different results: in fact, levels of IL-33 in sera of OA patients were found to be higher than in controls in their work [92]. The main findings of the IL-33 role in OA are reported in Table 2.

### 5.3. Role of S100B in Osteoarthritis

S100B appears to be also involved in OA pathogenesis. Zhu and colleagues have reported that OA human cartilage samples show augmented levels of S100B, TNF-α, and IL-1β, but more interesting is the described strong correlation between the levels of the alarmin and the two cytokines. Furthermore, the authors have demonstrated that human synovial fibroblasts from healthy knee patients with S100B overexpression and incubated in vitro with LPS increase their protein expression of TNF-α and IL-1β and mRNA and protein levels of fibroblast growth factor receptor 1 (FGFR1). This has been confirmed by the inverse results found with S100B knockdown through small interfering RNA (siRNA). Finally, the researchers have demonstrated that this alarmin acts through FGFR1 signalling; in fact, FGFR1 knockdown markedly reduced TNF-α and IL-1β levels in the conditioned medium of S100B-overexpressing fibroblasts, incubated with LPS [93]. RAGE is an important multiligand receptor whose role in OA has been recently demonstrated. As well as HMGB1, S100B demonstrates binding ability to this receptor. In fact, it has been demonstrated that human chondrocytes incubated with S100B and HMGB1 increase phosphorylated ratios of ERK-1/2 and p65, one of the NF-κB subunits, and protein levels of MMP-13, an enzyme with a key role in OA, for its digestive ability on type II collagen. All these findings are summarised in the Figure 4. Moreover, all these effects have been suppressed with soluble RAGE added to cell cultures. Additionally, Loeser and colleagues have reported that RAGE expression in normal chondrocytes increases with age; a greater number of immunostaining positive cells in old adults than in young, and a progression of positivity from cartilage surface into deep layers have been found [94]. The main evidence of the S100B role in OA is highlighted in Table 3.

Osteoarthritic joints show high levels of HMGB1, S100B, and IL-33; particularly, HMGB1 is increased in synovium and synovial fluid, as well as in cartilage. It is possible to assume that a primary injury against cartilage terminates with an initial discharge of HMGB1 that gives a start to a series of molecular events, turning all joint tissues into damaged. In fact, cell death leads to the passive release of HMGB1 in extracellular space, and it may act on adjacent cells, augmenting the expression of cytokines, chemokines, and enzymes and reducing the anabolic expression of collagen. Like in chondrocytes’ necrosis and/or apoptosis, extracellular matrix-derived molecules may induce active translocation of HMGB1 from the nucleus to the cytoplasm and then in the extracellular medium.

Afterwards, HMGB1-induced cytokines release, such as IL-1β and TNF-α, may determine augmented synthesis and release of HMGB1 [60,61,66]. This way, a positive feedback circuit may be generated with the initiation and amplification of sterile inflammation.

It is necessary to underline that OA arises from reiterated traumas, especially in joints with an unbalanced weight distribution. Hence, repeated impacts may set reiterated cell deaths and HMGB1 releases, constantly powering the circuit. These concepts are sustained by observations explaining that HMGB1 levels are correlated with histological grade of OA severity [60,61]. Finally, HMGB1 and cytokines (IL-1β) synergise in enhancing innate inflammation, further amplifying phlogosis [69].

High levels of HMGB1 in synovial fluid of OA joints have been found positively correlated with the radiological grade of severity (Kellgren-Lawrence system) [84] and clinical manifestations [72]. For all these reasons, alarmins, specifically HMGB1, may represent a completely new therapeutic target. In vitro, tests have shown that HMGB1 A-box, miR-129-5p, miR-142-3p, miR-140-5p, miR-93-5p, miR-107, chrysin, and glycyrrhizin can counter HMGB1 expression and improve the human osteoarthritic chondrocytes phenotype (rate of apoptosis and catabolic activities) [66,67,77,79,80,81,82,88]. Currently, disease-modifying drugs are not available, and physicians can only treat symptoms, trying to alleviate disability and partially delay surgical interventions.

Nowadays, osteoarthritis is increasingly considered an inflammatory disease with systemic involvement. In a mice model of osteoarthritis, high expression of AGEs and HMGB1 has been found in articular structures and has shown a positive correlation with histological damage in cartilage. Similarly, increased levels of different cytokines, in particular IL-1β, have been revealed in sera of the animals and a positive relationship with joint changes has been reported. Therefore, Kyostio-Moore and colleagues have demonstrated that OA in mice models is a disease with local and systemic inflammatory burden [95]. The same concept has been demonstrated in human patients since hsPCR and IL-6 levels were high in sera of OA patients [96,97].

Different papers have demonstrated the beneficial effects of HMGB1 blocking in animal models of OA. Aulin and colleagues have realised three different modalities of intra-articular injections of anti-HMGB1 in mice with OA induced by anterior cruciate ligament transection (ACLT) of the left knee joint. They have effectuated a single administration at the time of surgery (time 0) in the first group, two administrations at 0 and 2 weeks after the surgery in the second group, and two administrations at 2 and 4 weeks after the surgery in the third group. The results have been compared with other two groups: one injected with anti-TNF-α and one with saline solution (control). Based on the findings, anti-HMGB1 injections show similar results with anti-TNF-α and improved effects compared with the control. Additionally, late administration has not counteracted cartilage damage [68]. Last, Li and colleagues have demonstrated that intra-articular injections of anti-HMGB1 in TMJOA, induced by complete Freund’s adjuvant administration in rats, ameliorate articular inflammation and cartilage destruction [76].

Similar results have been obtained with intra-articular injections of antibodies neutralising both IL-33 and ST2 receptors in the murine model of OA; in fact, these treatments relieve pain and release of cartilage-degrading enzymes [91].

It is important to underline that interfering with HMGB1 treatment has revealed beneficial effects in different animal models and different ways of administration.

HMGB1 has also been demonstrated to play a role in the pathogenesis of murine arthritis models. Ostberg and colleagues have reported that nuclear retention of this alarmin through oxaliplatin intra-peritoneal treatment relieves collagen type II-induced arthritis in mice [98]. Kokkola and colleagues have illustrated an example of collagen-induced arthritis in mice and rats. The scholars have reported that polyclonal anti-HMGB1 antibodies and HMGB1 A-box intra-peritoneal therapies, reduce clinical disease activity and histologic damage [99]. The same results have been obtained by Schierbek and colleagues in their collagen-induced arthritis and spontaneous arthritis in mice with DNase type II and interferon type I receptor deficiencies, treated with intra-peritoneal polyclonal anti-HMGB1 antibodies [100].

Therefore, it is possible to assume that treatment against HMGB1, such as intra-articular injections of mAb, could be a future promising treatment option; specifically, it may be the first treatment interfering with the pathogenesis of the disease, partially blocking the course. Since it is a systemic inflammatory disease, it could be theoretically possible to counter the progression not only with local intra-articular administrations but also with systemic ones. To the best of our knowledge, in animal models of OA, several investigators have used local injections of HMGB1 antagonist drugs, unlike in other disease models, such as arthritis. However, these assumptions must be demonstrated with studies.

Furthermore, it would be necessary to demonstrate when the best clinical outcomes could be obtained. In fact, in animal models, early treatments lead to the best clinical results and it might be the same for human patients. The main concepts are summarised in Table 4.

## 6. Conclusions

In our review we have shown that alarmins play a pivotal role in OA pathogenesis; in fact, HMGB1, IL-33, and S100B have revealed higher expressions in OA chondrocytes compared with normal cartilage. Importantly, there are much more data available in the scientific literature for HMGB1 than for the other alarmins. Therefore, several studies are needed to deeply learn about their mechanisms in the disease cascade.

The new concept of OA, such as a local disease with a low grade of systemic inflammation, imposes new therapeutic horizons and also a different management perspective. 

Nowadays, great attention has been focused on HMGB1. Much evidence is available in the scientific literature about its role in different diseases. In particular, HMGB1 is involved in different autoimmune diseases, such as systemic lupus erythematous (SLE), idiopathic inflammatory myopathies, and rheumatoid arthritis. Hence, HMGB1 may be considered a new window into pathogenesis, focusing on thes discovery, monitoring, management, and treatment of different diseases other than OA.

## Figures and Tables

**Figure 1 ijms-24-12143-f001:**
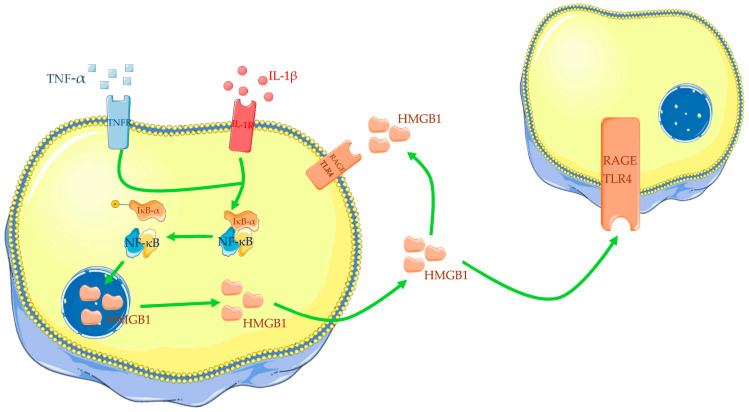
Cytokines, such as IL-1β and TNF-α, induce translocation of HMGB1 from the nucleus to the cytoplasm and then into the extracellular medium. HMGB1 could act in autocrine and paracrine ways. The Figure was drawn using Servier Medical Art, provided by Servier, licensed under a Creative Commons Attribution 3.0 unported license. It was partly modified and adapted to the tests.

**Figure 2 ijms-24-12143-f002:**
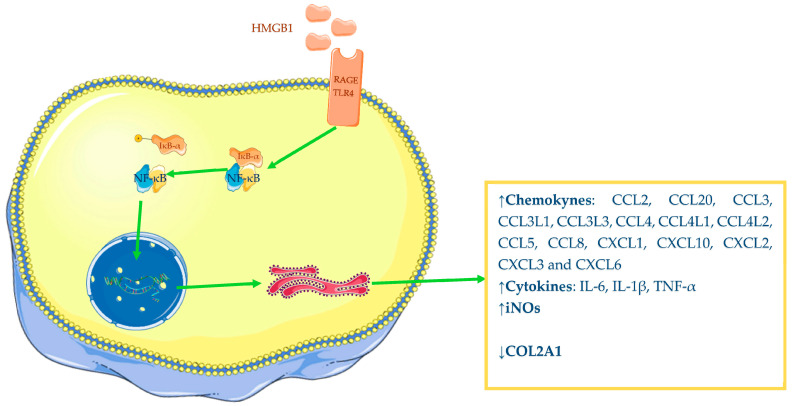
Pro-inflammatory power of HMGB1: increased synthesis of chemokine, cytokines, iNOS and decreased synthesis of COL2A1. The Figure was drawn using Servier Medical Art, provided by Servier, licensed under a Creative Commons Attribution 3.0 unported license. It was partly modified and adapted with tests.

**Figure 3 ijms-24-12143-f003:**
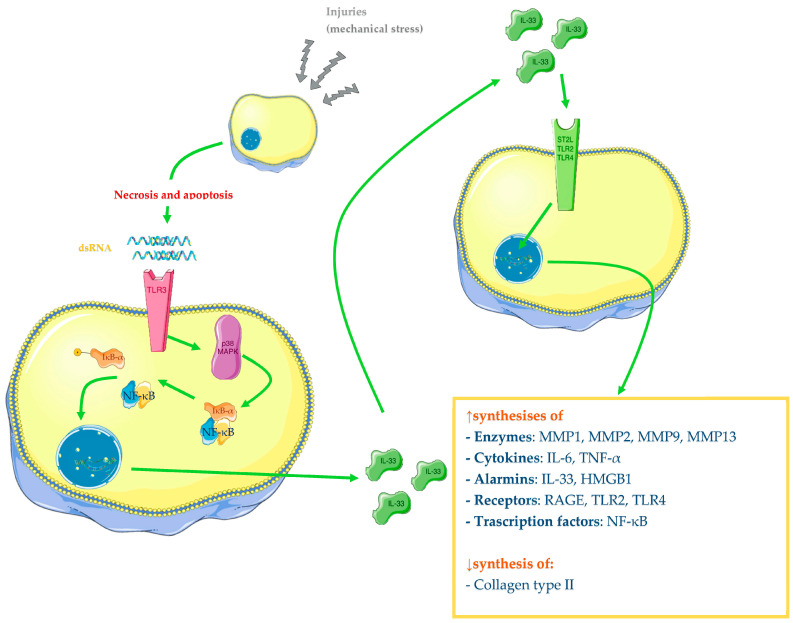
Role of IL-33 in the pathogenesis of OA in human chondrocytes. After ligand–receptor interaction, the MAP kinases system will be activated (only p38 is represented in the figure), leading to NF-κB activation and transcription of different genes (cytokines, enzymes, and other proteins), while the synthesis of collagen type II is downregulated. The Figure was drawn using Servier Medical Art, provided by Servier, licensed under a Creative Commons Attribution 3.0 unported license. It was partly modified and adapted with tests.

**Figure 4 ijms-24-12143-f004:**
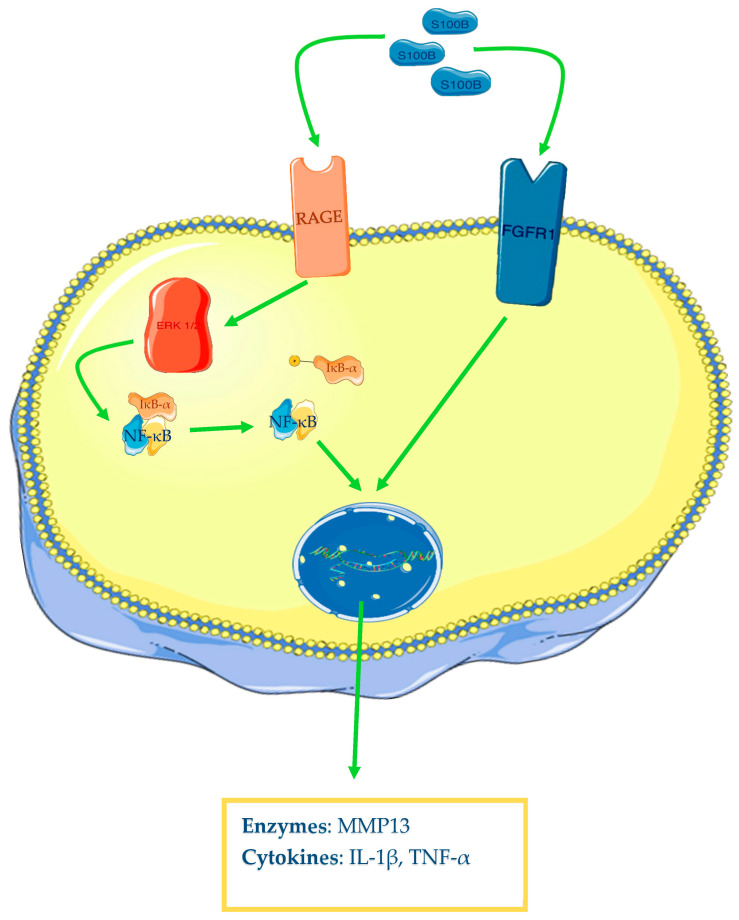
S100B and osteoarthritis. After ligand–receptor interaction, the MAP kinases system will be activated (only ERK1/2 is represented in the figure), leading to NF-κB activation and transcription of different genes (cytokines, enzymes) in chondrocytes. The Figure was drawn using Servier Medical Art, provided by Servier, licensed under a Creative Commons Attribution 3.0 unported license. It was partly modified and adapted with tests.

**Table 1 ijms-24-12143-t001:** HMGB1 and OA in vitro and in vivo results obtained through our research. The arrows stand for increased (↑) or decreased (↓) synthesis/expression.

Authors	Year	Cell Cultures	Biological Fluid	Outcomes
Terada et al. [60]	2011	(a) OA cartilage samples; (b) OA chondrocytes incubated with IL-1β and TNF-α; (c) OA chondrocytes incubated with HMGB1.		(a) ↑HMGB1 mRNA levels; gradually increased HMGB1 and RAGE protein expression and HMGB1 cytoplasmic positivity with progression of OARSI grade; (b) ↑HMGB1 translocation from the nucleus to the cytoplasm; (c) ↑TNF-α and IL-1β release in a dose-dependent manner for 12 h and 24 h, respectively, after stimulation.
Heinola et al. [61]	2010	Human bone marrow-derived mesenchymal stem cells differentiated in vitro into chondrocytes and then stimulated with TNF- α.		↑HMGB1 expression and cytoplasmic positivity; gradual increase of HMGB1 positivity with histological severity of OA.
Rosenberg et al. [62]	2017	(a) Normal human articular chondrocytes vs. OA chondrocytes; cartilage tissues from (b) OA knee vs. (b) OA hip.		(a)↑ HMGB1, RAGE, S1008, and S1009 protein levels, and HMGB1, RAGE, and S1008 mRNA expression in OA chondrocytes; HMGB1, RAGE, S1008, and S1009 protein levels, and HMGB1, RAGE, and S1008 mRNA expression in OA knee in (b) > (a).
Amin et al. [63]	2014	(a) Cartilage of the deep region from OA knees; (b) cartilage from OA knees incubated with HMGB1.		(a) ↑HMGB1 positivity compared with normal, in the cytoplasm and extracellular milieu; spontaneous release of HMGB1 from OA cartilage in ex vivo conditions; (b) ↑mRNA of CCL2, CCL20, CCL3, CCL3L1, CCL3L3, CCL4, CCL4L1, CCL4L2, CCL5, CCL8, CXCL1, CXCL10, CXCL2, CXCL3 and CXCL6, NF-κB1, NF-κB2; ↑mRNA and protein levels of IL-8, iNOs; HMGB1 induces paracrine effects on OA cartilage.
Wagner et al. [64]	2021	(a) OA cartilage samples; (b) chondrogenic progenitor cells incubated with different concentrations of HMGB1.		(a) ↑HMGB1 protein level; HMGB1 positivity in the cytoplasm and extracellular medium; (b) ↑Migration rate (time- and dose-dependent).
Hwang et al. [65]	2018	Cartilage from OA knee		↑Nucleus to cytoplasm translocation and extracellular release of HMGB1; ↑p-mTOR; ↑Bcl-2/Beclin-1; ↓HMGB1/Beclin-1.
Fu et al. [66]	2016	(a) Normal human chondrocytes pretreated with IL-1β and (b) then with HMGB1 A-box overexpression.		(a) ↑ mRNA and protein levels of HMGB1, ADAMTS-4, ADAMTS-5, and TLR4, ↑secretion of MMP-1, MMP-3 and MMP-9, ↑mRNA levels iNO and COX-2; ↑production NO and PGE2, ↑p-p65 levels; (b) ↓mRNA and protein levels of HMGB1, ADAMTS-4, ADAMTS-5 and TLR4, ↓secretion of MMP-1, MMP-3 and MMP-9, ↓mRNA levels iNO and COX-2; ↓production NO and PGE2, ↓p-p65 levels.
Zhou et al. [67]	2021	(a) Chondrocytes from OA patients; (b) chondrocytes from OA patients incubated with IL-1β and glycyrrhizin.	(c) synovial fluid from OA patients	(a) ↑HMGB1 protein levels; (b) ↑cell viability, ↓mRNA and protein levels of HMGB1, TNF-α, IL-6, MMP-1, MMP-3, and MMP-13; ↓PGE2 and NO production (c) ↑HMGB1 compared with normal.
Aulin et al. [68]	2020	OA knees chondrocytes incubated with dsHMGB1		↑IL-6, IL-8 protein levels and ↓mRNA of COL2A1.
Ding et al. [69]	2017	Normal human chondrocytes from weight-bearing joints incubated with IL-1β and (a) HMGB1 or (b) FN-f.		MMP-3, MMP-13, ADAMTS-5, ADAM-8, and iNOS synthesis in (a) > (b).
Sun et al. [70]	2016	Synovial tissues from OA knees		↑mRNA and protein levels of HMGB1 and RAGE; HMGB1 and RAGE expression is positively correlated with X-ray severity.
García-Arnandis et al. [71]	2010	(a) OA synoviocytes; OA synoviocytes incubated with HMGB1 (b) with or (c) without IL-1β.		(a) ↑HMGB1 expression and cytoplasmic positivity compared with normal (statistical significance not reached); mRNA and protein expression IL-6, IL-8, CCL2, CCL20, MMP1, MMP3 in (b) > (c); mRNA and protein expression MMP13 in (b) > (c) but statistical significance not reached; p-ERK1/2, p-p38 and p-Akt in (b) > (c).
Ke et al. [72]	2015	(a) Synovium from knee OA patients.	(b) Synovial fluid from OA patients	(a) ↑HMGB1 expression and cytoplasmic positivity compared with normal; (a,b) ↑HMGB1 levels compared with normal; (b) ↑HMGB1 in the radiological KL 2/3 group compared with KL 4; HMGB1 positively correlated with OA clinical findings: pain, synovitis, and daily activities.
Feng et al. [73]	2016	Synovial membrane from TMJOA patients.		↑mRNA and protein levels HMGB1.
Wähämaa et al. [74]	2011	Synovial fibroblast from OA patients incubated with HMGB1 in complex with (a) LPS or (b) IL-1β.		(a,b)↑TNF-α, IL-6, IL-8 and MMP-3; cytokine production in (a) > (b); (a) HMGB1-LPS complex interacts with TLR4; (b) HMGB1- IL-1β complex interacts with IL-1RI.
Hreggvidsdottir et al. [75]	2009	Synovial fibroblasts from OA patients incubated with (a) HMGB1 alone, (b) IL-1β alone, or with (c) HMGB1 and IL-1β.		IL-6 levels in (c) > (a,b)
Li et al. [76]	2022	Human synovial fibroblasts from OA temporomandibular joint treated with HMGB1.		↑MMP13, ADAMTS5, IL-1β, IL-6 and p-NF-κB p65; ↑nuclear localisation of p-NF-κB p65.
Qiu et al. [77]	2021	Chondrocytes incubated with IL-1β and (a) exosomes poor in miR-129-5p compared with (b) exosomes rich in miR-129-5p.	(c) Synovial fluid from OA patients	Protein levels of HMGB1, TLR4, p-NF-κB, COX2, iNOS and MMP13 in (a) > (b); apoptosis rate in (a) > (b); collagen 2 in (a) < (b); (c) ↑HMGB1 and IL-1β and ↓miR-129-5p; miR-129-5p levels are negatively correlated with HMGB1.
Jiang et al. [78]	2017	(a) OA human chondrocytes; (b) OA human chondrocytes incubated with IL-1β and JQ1.		(a) ↑mRNA and protein levels BRD4; BRD4 levels positively correlated with histological grade of OA (OARSI); (b) ↓mRNA and protein levels of HMGB1, IL-6, TNF-α, MMP3, MMP9, and MMP13; ↓HMGB1 from the nucleus to cytoplasm translocation; ↓p65 from the cytoplasm to the nucleus.
Gao et al. [79]	2019	(a) Cartilage from OA knee or hip; (b) chondrocyte from OA patients with MCM3AP-AS1 overexpression; (c) chondrocyte from OA patients with miR-142-3p overexpression; (d) chondrocyte from OA patients incubated with crescent doses of LPS (0-2000 ng/mL) for 24 h.		(a) ↑MCM3AP-AS1 (b) ↑HMGB1 mRNA and protein levels; ↑apoptotic rate; (c) ↓HMGB1 mRNA and protein levels; ↓apoptotic rate; (d) ↑MCM3AP-AS1 and HMGB1 mRNA; ↓miR-142-3p.
Wang et al. [80]	2020	(a) Cartilage from OA knee; (b) normal human chondrocytes C28/I2 incubated with IL-1β and then transfected with miR-140-5p mimics.		(a) ↓miR-140-5 and ↑HMGB1 mRNA levels; (b) ↓HMGB1, p-PI3K, p-AKT, MMP-1, MMP-3, TNF-α and IL-6 protein levels; ↓apoptosis rate; ↑cell viability (CCK-8).
Meng et al. [81]	2020	(a) Human normal chondrocytes C28/I2 stimulated with LPS and (b) then with si-PVT1; (c) human normal chondrocytes C28/I2 stimulated with LPS and then with miR-93-5p	(d) Serum from OA patients	(a) ↑PVT1, ↓miR-93-5p, ↑apoptosis rate, IL-6, IL-1β, TNF-α, MMP13, p-p65, TLR4, and p-IκB-α expression; (b) ↓PVT1, ↑miR-93-5p, ↓apoptosis rate, IL-6, IL-1β, TNF-α, MMP13, p-p65, TLR4, and p-IκB-α expression; (c) ↓mRNA and protein HMGB1; (d) ↑PVT1 and ↓miR-93-5p compared with controls.
Lin et al. [82]	2019	(a) OA articular cartilage specimens transfected with miR-107 mimics; (b) OA chondrocytes exposed to hyperbaric oxygen treatment.		(a) ↓HMGB1; (b) ↑miR-170, ↓HMGB1, TLR2, TLR4, RAGE and iNOS mRNA and protein levels, ↓HMGB1, MMP-9, and MMP- 13 extracellular release, ↓p38 MAPK, ERK and JNK phosphorylation, ↑IκBα protein synthesis.
Feng et al. [83]	2021	(a) Human synovial fibroblasts from TMJOA stimulated with HMGB1; (b) human umbilical vein endothelial cells (HUVEC) incubated with conditioned medium from fibroblasts stimulated with HMGB1; (c) human umbilical vein endothelial cells (HUVEC) incubated with conditioned medium from fibroblasts stimulated with HMGB1 and anti-VEGF.		(a) ↑VEGF, HIF-1α, p-Erk, p-JNK; (b) ↑migration and tube formation of HUVEC; (c) ↓migration and tube formation of HUVEC.
Zhan-Chun et al. [84]	2011		Synovial fluid from OA patients	↑HMGB1 compared with controls; HMGB1 levels are positively correlated with X-ray severity.
Aulin et al. [85]	2022		Synovial fluid from patients with recent or old knee injuries, OA knee, and a healthy knee	Expressions of HMGB1 in recent injury > old injury > OA; HMGB1 is associated with cartilage biomarkers.
Ding et al. [86]	2020		Synovial fluid from patients with acute or chronic anterior cruciate ligament (ACL) injuries	HMGB1 in chronic group > acute group (p=0,075 statistical significance not reached).
García-Arnandis et al. [87]	2010	OA patients’ synoviocytes incubated with IL-1β and overexpression of haem oxygenase-1.		↓mRNA and protein levels of HMGB1, MMP-1 and MMP-3; ↓ from the nucleus to cytoplasm translocation of HMGB1.
Zhang et al. [88]	2018	(a) Human chondrocytes pretreated with IL-1β and (b) then incubated with chrysin.		(a) ↑HMGB1, MMP-13, collagenase, and IL-6; ↓COL2A1; ↑apoptotic rate (b) ↓HMGB1, MMP-13, collagenase, and IL-6; ↑COL2A1; ↓apoptotic rate.

**Table 2 ijms-24-12143-t002:** IL-33 and OA in vitro results were obtained through our research. The arrows stand for increased (↑) or decreased (↓) synthesis/expression.

Authors	Year	Cell Cultures	Biological Fluid	Outcomes
Rai et al. [89]	2022	(a) Osteoarthritic human chondrocytes: (b) OA knee and (c) OA hip chondrocytes; (d) NHAC incubated with IL-33; (e) NHAC incubated with IL-37; (f) NHAC incubated with IL-37 followed by IL-33, LPS, or rHMGB1 and (g) NHAC incubated only with IL-33, LPS or rHMGB1.		(a) ↑mRNA of IL-33, IL-37, TLR2, TLR4, NF-κB, IL-6, TNF-α, MMP2, and MMP9; ↑protein expression IL-37, TLR2, TLR4, NF-κB, IL-6, TNF-α, MMP2, and MMP9; mRNA and protein levels of IL-33, TLR2, TLR4, NF-κB, MMP2, and MMP9 in (b) > (c); protein level of IL-37 in (c) > (b); (d) ↑mRNA of IL-33, TNF-α, IL-6, TLR2, TLR4, MMP2, MMP9, NF-κB, HMGB1, RAGE, and ↑M1 macrophage; (e) ↓mRNA levels of IL-37, TLR2, TLR4, IL-6, TNF-α, NF-κB, MMP, MMP9, RAGE and HMGB1; ↓TLR2, TLR4, IL-6, TNF-α, NF-κB, MMP2, MMP9, RAGE, and HMGB1 mRNA levels in (f) < (g).
Li et al. [90]	2017	(a) OA knee cartilage samples: (c) chondrocytes of weight-bearing areas; (d) normal human primary chondrocytes incubated with supernatant from healthy and (e) injured cartilage lysates; (f) normal human primary chondrocytes incubated with supernatant from injured cartilage lysates with and (g) without RNase A; (h) normal human primary chondrocytes incubated with supernatant from injured cartilage lysates with TLR3 silencing and (i) with TLR7 knocking down; (l) IL-33-stimulated human chondrocytes in vitro; human chondrocytes incubated with commercial dsRNA analogue poly(I:C) and (m) with or (n) without IL-33 siRNA.	(b) Synovial fluid from OA patients	(a)↑mRNA and protein levels of IL-33, MMP1 and MMP13; (b)↑protein levels of IL-33, MMP1 and MMP13; (c)↑mRNA levels of IL-33, MMP1 and MMP13; ↑mRNA and protein levels of IL-33, MMP1 and MMP13 and ↓ collagen II in (e) > (d); ↑mRNA and protein levels of IL-33, MMP1 and MMP13, and ↓collagen II in (g) > (f); ↑mRNA and protein levels of IL-33, MMP1 and MMP13, and ↓collagen II in (i) > (h); (l) ↑mRNA and protein levels of MMP1 and MMP13 and ↓collagen II; mRNA of MMP1 and MMP13 in (n) > (m).
He et al. [91]	2020	(a) OA human chondrocytes	(b) Serum and synovial fluid of OA patients	(a) ↑mRNA and protein levels of IL-33 and ST2; (b) ↑levels of IL-33 only in synovial fluid.
Hong et al. [92]	2011	Serum of OA patients		↑Levels of IL-33

**Table 3 ijms-24-12143-t003:** S100B and OA in vitro results obtained through our research. The arrows stand for increased (↑) or decreased (↓) synthesis/expression.

Authors	Year	Cell Cultures	Biological Fluid	Outcomes
Zhu et al. [93]	2018	(a) OA cartilage samples: human synovial fibroblast from the normal knee (b) transfected with S100B overexpression or (c) knockdown siRNA and incubated with LPS; (d) human synovial fibroblast transfected with S100B overexpression, incubated with LPS, and exposed to FGFR1 siRNA.		(a) ↑S100B, TNF-α and IL-1β levels; strong correlation between S100B levels and TNF-α and IL-1β expression; (b) ↑TNF-α and IL-1β protein levels and FGFR1 mRNA and protein expression in medium; (c) ↓TNF-α and IL-1β levels and FGFR1 mRNA and protein expression in medium; (d)↓TNF-α and IL-1β levels.
Loeser et al. [94]	2005	(a) chondrocytes incubated with HMGB1 and S100B; (b) chondrocytes incubated with HMGB1 and S100B and soluble RAGE.		(a) ↑p-ERK-1/2, p-p65 and MMP-13 protein levels; (b) ↓p-ERK-1/2, p-p65 and MMP-13 protein levels.

**Table 4 ijms-24-12143-t004:** Key points.

Key Points
HMGB1, S100B, and IL-33 levels are increased in OA joints compared with normal joints. These three alarmins promote pro-inflammatory and catabolic phenotypes in OA chondrocytes. There is a greater amount of available data for HMGB1 than for S100B and IL-33. These three alarmins act through different signalling, the most important being NF-κB –mediated signalling. There are several axes between HMGB1 and RNAs influencing OA initiation and progression. Treatment against HMGB1 has revealed improved effects on the OA phenotype of joint cells, in vitro and in vivo models.

## Data Availability

Data are taken from previously published articles since this is a review of the current literature to date.

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
