# Peer review of "The Role of Alarmins in Osteoarthritis Pathogenesis: HMGB1, S100B and IL-33"

_ijms, 2023, doi:10.3390/ijms241512143_

Round 1
Reviewer 1 Report
Aim of the present revew is to evalue the roles of the three alarmins HMGB1,IL33 and SI00B in the pathogenesis of OA.These objectives of the revew have been obtained through the exhaustive examination of the literature,a very useful insert of the experiments in the tables,very clear figures and language,resulting a very useful paper for the researchers on the field of OA.
Author Response
Dear reviewer, I revised the paper accordingly to your suggestions.
Giuseppe Murdaca

Reviewer 2 Report
It is well reviewed the three HMGB1, S100B and IL-33 alarmins in OA.
Please clarify at Aim of the study that why HMGB1, S100B and IL-33 alarmins from the other alarmin proteins were chosen for this review.
minor editing of English language is required.
Author Response

(The authors gave the same response as above.)

Reviewer 3 Report
Thank you for the nice review concerning the role of alarmins in osteoarthritis.
I have the following remarks:
L31 Please add : chondrophyte and osteophyte formation. Is angiogenesis really a characteristic? Please reconsider
L41 Prevalence of processes is incorrect. Please adapt
L59 Patterns instead of pattern. PAMPs is prural
L69 the cell. please add ´the´
L71 cell release instead of releasing
L74 it chemoattracts
L108 ST2V is expressed in the ......
L126 The aims of the study (a) to (e ) are not clearly followed in the text. Please clarify the text accordingly or adapt the definition of the aims.
L130-132. Please rephrase
L134 performed instead of effectuated
L137 arguments of the study meaning??? Please rephrase
L141 Results and discussion. There is considerable overlap and redundancy between the text and tables. Please condense to improve readibility
L201 it is plausible to speculate....? Recruitement of CPC into a damaged ECM with apoptic or necrotic chondrocyte is a very unlikely event. Please adapt accordingly
L209 please include ligaments and menisci
L218 demonstrated instead of explained
L280 has confirmed an improvement activity? Please rephrase
L317 Hyperbaric oxygen (HBO) therapy. Please adapt for readibility
L336 Kellgren-Lawrence (KL)
L366 Various instead of Diverse
L389 starts instead of gets starts to
L462-463 IL-33 increases mRNA levels of IL-33? Autostimulation? Please explain or adapt
L474 researchers instead of scholars?
L497 Study of Hong [92] is concerned with rheumatoid arthritis not OA. Hence the difference?
L544 receptor instead of receptors
L672-683 This section seems completely disconnected from the aims of the study and alarmins & OA data presented. Please address this to bring the conclusion section in line with your study aims
L690 There is HMGB1/miRNA axes. If you mean axes in plural, a proposal could be: There are several ....axes influencing OA initiation and progression
Please see my previous remarks
Author Response

(The authors gave the same response as above.)

Reviewer 4 Report
excellent review, with details, well - presented to the reader with figures, tables and fluent use of english.
I have noticed some references that according to my opinion should be replaced in the text. For example line 365. I found very interesting this particular information, regarding the potential effect of bleeding in the cascade of inflammation in the joint. However, the paper that corresponds to reference 87, is not refering in bleeding situation but to an experiment of basic science. As writen in your article, your opinion (or your colclusion, if you want) that is not incorrect of course, should not be presented as the main idea of the referrence number 87. Therefore, replacement of the reference a few lines above, would be better and the reader will not be confused.
Author Response

(The authors gave the same response as above.)
